# Formation of Abasic Oligomers in Nonenzymatic Polymerization of Canonical Nucleotides

**DOI:** 10.3390/life9030057

**Published:** 2019-07-04

**Authors:** Chaitanya V. Mungi, Niraja V. Bapat, Yayoi Hongo, Sudha Rajamani

**Affiliations:** 1Indian Institute of Science Education and Research (IISER), Pune 411008, Maharashtra, India; 2ELSI, Tokyo-Tech (Earth-Life Science Institute, Tokyo Institute of Technology), 2-12-1, Ookayama, Meguro-ku, Tokyo 152-8550, Japan; 3OIST, Okinawa Institute of Science and Technology Graduate University, 1919-1 Tancha, Okinawa 904-0412, Japan

**Keywords:** prebiotic polymerization, nucleotide oligomerization, abasic oligomers, nucleotide stability, stability as a selection pressure, dry-wet cycles

## Abstract

Polymerization of nucleotides under prebiotically plausible conditions has been a focus of several origins of life studies. Non-activated nucleotides have been shown to undergo polymerization under geothermal conditions when subjected to dry-wet cycles. They do so by a mechanism similar to acid-catalyzed ester-bond formation. However, one study showed that the low pH of these reactions resulted in predominantly depurination, thereby resulting in the formation of abasic sites in the oligomers. In this study, we aimed to systematically characterize the nature of the oligomers that resulted in reactions that involved one or more of the canonical ribonucleotides. All the reactions analyzed showed the presence of abasic oligomers, with purine nucleotides being affected the most due to deglycosylation. Even in the reactions that contained nucleotide mixtures, the presence of abasic oligomers was detected, which suggested that information transfer would be severely hampered due to losing the capacity to base pair via H-bonds. Importantly, the stability of the *N*-glycosidic linkage, under conditions used for dry-wet cycling, was also determined. Results from this study further strengthen the hypothesis that chemical evolution in a pre-RNA World would have been vital for the evolution of informational molecules of an RNA World. This is evident in the high degree of instability displayed by *N*-glycosidic bonds of canonical purine ribonucleotides under the same geothermal conditions that otherwise readily favors polymerization. Significantly, the resultant product characterization in the reactions concerned underscores the difficulty associated with analyzing complex prebiotically relevant reactions due to inherent limitation of current analytical methods.

## 1. Introduction

Polymerization of monomers under prebiotically plausible conditions would have been an essential event during the origin of life on Earth. Aligning with the RNA World hypothesis, most efforts in the past were targeted towards making RNA molecules using nonenzymatic polymerization methods. Several studies have looked at polymerization of imidazole ‘activated’ nucleotides. However, both the availability of these monomers in concentrated amounts and their polymerization process under early Earth conditions remains uncertain [1]. A few studies have looked at the polymerization of non-activated nucleotides and have reported the formation of RNA-like polymers by subjecting these monomers to dehydration-rehydration (DH-RH) cycles in the presence of lipids [2,3]. These DH-RH reactions were carried out under simulated volcanic geothermal conditions, a niche thought to have been prevalent on the prebiotic Earth. The chemical and thermal fluxes associated with this niche are considered to have facilitated pertinent prebiotic processes relevant to the emergence of life [4,5,6]. Detailed biochemical characterization of the reaction products from lipid-assisted polymerization of adenosine 5′ monophosphate (5′-AMP), suggested the presence of abasic sites in the resultant oligomers [7]. Loss of base resulted from the cleavage of the *N*-glycosidic bond, as the reactions were carried out at low pH. Depurination has been previously studied in the biological context in which abasic RNA was found to be significantly more stable than abasic DNA [8]. This study looked at the rate of strand cleavage by β,δ-elimination and 2′,3′-cyclophosphate formation and reported a 15-fold reduction in cleavage when abasic sites were present in RNA as compared to in DNA. In particular, the mechanism for loss of base has been predominantly studied in DNA-based systems due to their biological relevance. Loss of base was found to be favored at low pH, especially below pH 2.4, which corresponded to the pKa of the nucleobases that were studied [9]. Another study reported a steep increase in depurination with increased temperatures [10]. The sigmoidal curve obtained for the rate of depurination vs. temperature showed a dramatic increase in slope due to the loss of base above 85 °C.

Thereafter, a systematic kinetic analysis of the deglycosylation reaction was carried out for modified and canonical nucleosides [11]. Acidic conditions lowered the enthalpic activation parameter (ΔH) of deglycosylation, thus enhancing the ability of the leaving group and resulting in the loss of base, especially in the case of purines. The pKa of the nucleobases was found to be correlated with the stability of the glycosidic bond; bases with higher pKa (more basicity) also had higher ΔH, which resulted in lower rate constants for deglycosylation. Specifically, purine deoxyribonucleosides had half-lives (t_1/2_) close to 15 min, whereas their ribonucleoside counterparts showed half-lives of about 7–8 days under low pH (0.1 M HCl) and physiological temperatures (37 °C). The study also reported half-lives at higher temperatures of 50 °C, and the t_1/2_ for deoxyadenosine decreased further to only 3.2 min. In particular, the main claim of this study was that the canonical bases have the lowest rates of deglycosylation at physiological pH when compared to other modified bases. It has been argued that the stability of *N*-glycosidic linkages would have been one of the pertinent selection pressures that allowed for the refining of the genetic alphabet during transition from the RNA World(s) to a DNA-based system, allowing for efficient encoding of information [12,13].

Given the aforementioned data and our observation of depurination in polymerization reactions involving 5′-AMP [7], we decided to systematically characterize the products resulting from other contemporary RNA nucleotides when used as starting monomers. Towards this extent, we have already reported the preliminary results from reactions involving 5′-GMP, 5′-CMP, and 5′-UMP (and combinations thereof). These monomer-based reactions also resulted in the formation of oligomers (Appendix A) [7], but their exact biochemical nature was not ascertained. In a relevant study, native pyrimidines were observed to have long half-lives (t_1/2_ of over 400 days) at pH 1 and 37 °C [14]. This therefore makes pyrimidine-based monomers better candidates for studying acid-catalyzed polymerization due to their higher glycosidic bond stability. Furthermore, the possibility of base pairing has been previously demonstrated in oligomers formed in DH-RH reactions [15], which was argued to positively impact the yield of ‘intact’ oligomers. To study this in greater detail, mixtures of nucleotides capable of base pairing (i.e., AMP + UMP and GMP + CMP), as well as a mixture of all four nucleotides, were subjected to DH-RH reaction conditions at low pH. All the resultant products were analyzed using mass spectroscopy to delineate the oligomers formed in various reactions. It is pertinent to consider the complexity that could emerge in the resultant products, as such ‘mixed’ nucleotide reactions would result in heteropolymers containing more than one type of nucleobase. This complexity is reflective of how processes would have progressed in a heterogeneous prebiotic soup. In this kind of a scenario, multiple reactants and reactions would have been affected concomitantly, with the product distributions reflecting the varying reactivity and availability of the reactants involved. Similar trends were also observed in our reactions, emphasizing the importance of factoring in molecular complexity while analyzing prebiotically pertinent reactions.

## 2. Materials and Methods

### 2.1. Materials

Adenosine 5′-monophosphate (AMP), uridine 5′-monophosphate (UMP), guanosine 5′-monophosphate (GMP), cytidine 5′-monophosphate (CMP), and ribose 5′-monophosphate (rMP) were purchased as disodium salts from Sigma-Aldrich (Bangalore, India) and used without further purification. The phospholipid, 1-palmitoyl-2-oleoyl-sn-glycero-3-phosphocholine (POPC), was purchased from Avanti Polar Lipids Inc. (Alabaster, AL, USA). All other reagents used were of analytical grade and purchased from Sigma-Aldrich (Bangalore, India).

### 2.2. Methods

#### 2.2.1. Dehydration-Rehydration Cycles

Oligomerization reactions were carried out in the same set up described previously [7]. The parameters used for DH-RH cycles were selected based on previous experiments. Seven DH-RH cycles were carried out at 90 °C with 1 h of drying time. The pH of the reaction was lowered using H_2_SO_4_ and Milli-Q water was used as rehydrating agent for subsequent cycles. In order to check the polymerization of nucleotides, reactions were carried out with a ratio of 1:5 of lipid:nucleotide. Additionally, binary mixtures of nucleotides capable of base pairing, i.e., AMP + UMP and GMP + CMP, were also subjected to DH-RH cycles using a 1:1 ratio of both nucleotides (e.g., 2.5 mM AMP + 2.5 mM UMP). Finally, a reaction with all four nucleotides in equal ratio (1.25 mM each) was also carried out under the same reaction conditions.

#### 2.2.2. Analysis of Deglycosylation

Deglycosylation reaction analysis was carried out mainly for AMP, as it has been reported to have the highest *N*-glycosidic bond stability amongst the canonical purines [11]. The AMP solution was maintained at pH 2 using H_2_SO_4_ and dried at 90 °C. Separate reactions vials were used for individual time points (as detailed in Section 3), and the reactions were analyzed for the loss of base. Deglycosylation reactions were also carried out without dehydration of the starting reaction mixture by heating the solutions in closed tubes. This was done in order to prevent the oligomerization that also takes place under these conditions. These samples were then analyzed by HPLC for evaluating the breakdown in each sample, and the percentage depurination was estimated as follows: (area of breakdown peak/area of monomer peak) × 100. The time required for the breakdown of *N*-glycosidic linkage was calculated by plotting the percentage depurination against time.

#### 2.2.3. HPLC Analysis

Chromatography was performed using an Agilent 1260 chromatography system (Agilent Technologies, Santa Clara, CA, USA) and DNAPac PA200 column (4 × 250 mm) from Dionex (now Thermo Scientific, Sunnyvale, CA, USA). Samples were analyzed with a linear gradient of NaClO_4_ in 2 mM Tris buffer at pH 8 using a flow rate of 1 mL/min. All solvents, purchased from Sigma-Aldrich (Bangalore, India), were of HPLC-grade and used after filtering through a 0.22 μm nylon filter followed by degassing. Samples were detected using a high-sensitivity flow cell (60 mm path length) in a diode array detector. As some reactions contained more than one type of monomer, each nucleotide was injected separately as a control, and the retention time of the respective peaks was noted. In some cases, more than one monomer peak eluted together (AMP and GMP) due to the specificity of column for phosphate groups and not necessarily for the nitrogenous bases. This aspect of the column chemistry did not allow for further optimization of peak elution profiles. Since this column offered the best single-nucleotide resolution, we performed qualitative analysis using this column despite the aforementioned limitation of this technique.

#### 2.2.4. Mass Analysis

Detailed mass analysis of nucleotide mixtures was carried out at the Earth Life Science Institute (ELSI), Tokyo, Japan. Samples were lyophilized and shipped to ELSI at Tokyo Institute of Technology, and further mass analysis was carried out there using the Acquity UPLC+ system from Waters Corp. (Milford, MA, USA) with a CORTCES UPLC C18 column (1.6 μm, 2.1 × 50 mm) using a water/acetonitrile gradient containing 0.1% trifluoroacetic acid. Mass determination was carried out in the positive ion mode with XEVO G2-XS QTof Mass Spectrometry using MassLynx ver. 4.1 (Waters Corp.)

## 3. Results and Discussion

### 3.1. Polymerization of Canonical Nucleotides

Initially, DH-RH reactions were performed with individual nucleotides. The reaction mechanism relies on the protonation of the phosphate group and subsequent nucleophilic attack of the 2′/3′ OH of a neighboring monomer, which results in a phosphodiester bond. This proposed mechanism is independent of the nucleobase. Given this, in principle, all four nucleotides should effectively polymerize by the aforementioned acid-catalyzed esterification mechanism. We replicated the previously reported results and observed similar HPLC chromatograms for all nucleotides (Appendix A). Purine reactions showed greater polymerization; however, peaks observed in the dead volume indicated loss of nucleobases and formation of abasic oligomers.

This was further confirmed by performing mass analysis of these samples. Mass spectrometry (MS) was performed on individual nucleotide controls and on the reaction samples. Controls for all nucleotides showed the expected mass for the monomer. Additionally, mass numbers corresponding to stacked oligomers, which could result from association of molecules due to ionization, were also observed (Appendix A). Fragmentation of the monomer was also observed during MS as the control nucleotide samples showed mass numbers corresponding to free bases, ribose 5′-phosphate (rMP), and, in some cases, phosphate (iP) as well. Due to the possibility of such fragmentation during MS acquisition, loss of base was considered to have taken place during the reaction only if it was observed in both HPLC as well as the MS analysis.

Analysis of the reaction samples showed the presence of oligomers in all the reactions. The mass numbers obtained from this analysis are summarized in Table 1. Purine nucleotide-based reactions showed the presence of mass numbers corresponding to the respective free bases, monomers, abasic dimers, and abasic trimers. For the reactions containing pyrimidine nucleotides (UMP and CMP), mass numbers corresponding to the monomer and intact dimers were observed in both the cases. Notably, these were the only two reactions that showed the presence of intact dimers, indicating greater stability of the glycosidic bond in the pyrimidine nucleotides. However, mass numbers corresponding to free bases and abasic dimers were also observed in the pyrimidine-based reactions. Nonetheless, this did not corroborate with the observations from the corresponding HPLC analysis wherein no breakdown peaks were observed for these reactions. The presence of free base in the mass spectrum can thus be attributed to fragmentation during the ionization, especially in the case of reactions involving only pyrimidine nucleotides. Mass spectra of pyrimidine control samples also showed the presence of free bases, confirming that they potentially resulted from fragmentation during MS data acquisition. Based on the combined evidence from HPLC and MS, we can infer that the glycosidic bond cleavage in pyrimidine reactions might not have occurred during the DH-RH cycles. As stated in the introduction section, the rate of deglycosylation is variable amongst nucleobases, with purines being the most prone. Our results corroborate this observation as deglycosylation was predominantly observed in the purine reactions.

### 3.2. DH-RH Reactions for Nucleotide Mixtures Capable of Hydrogen Bonding

Subsequently, reactions were also carried out under aforementioned conditions with nucleotide mixtures capable of hydrogen bonding (H-bonding). In a previous study, it was shown that the oligomers that resulted during the DH-RH reactions of AMP and UMP could hydrogen bond as assessed by the hyperchromicity analysis of the reaction products [15]. However, results obtained from our experiments suggest that the formation of abasic oligomers from purine nucleotides would potentially decrease the number of H-bonds that can be formed for a given length of oligomer. Importantly, it is well known that the canonical nucleotide monomers do not form H-bonded pairs by themselves in solution [16]. Nonetheless, in order to check whether starting with a mixture of base-pairing nucleotides might indeed positively affect the outcome of oligomer formation, reactions were carried out with binary mixtures of nucleotides (AMP + UMP and GMP + CMP) and the samples were analyzed. Base-specific separation of monomers and/or oligomers could not always be achieved using the HPLC as in previous reports [7] (Appendix A). Both reactions showed the presence of dead volume peaks, which typically correspond to free bases. However, it was difficult to determine whether purines or pyrimidines were being predominantly lost. However, results from the individual nucleotide reactions suggested that the loss of purines might be the major contributor to the observed HPLC breakdown peaks in these binary reaction mixtures. Subsequent characterization was, therefore, carried out by mass analysis of the resultant oligomers.

Table 2 summarizes the mass numbers observed in the mass spectrometric analysis of A + U and G + C reactions. Though exact abundances of the peaks were different in these reactions, all these reaction mixtures were comprised of abasic oligomers with some purines and pyrimidines still left intact. Oligomers found in these reactions were similar to those observed in purine-only reactions that contained multiple abasic sites with a single intact base (as depicted in Appendix A). Intact dimers were observed only in the G + C reaction but with relatively poor abundance. Since quantitative MS analysis was not performed, it is beyond the scope of this work to comment on the yields of oligomers based just on the corresponding mass abundances. Peaks observed in the dimer and trimer populations mainly consisted of abasic oligomers, often with only one intact base remaining. Completely abasic oligomers (such as rMP-rMP dimers, Appendix A) were not observed in these reactions possibly due to ligation of such products with other species, resulting in higher oligomers (e.g., formation of a species like AMP-rMP-rMP, etc., Appendix A). The loss of base could be predominantly occurring during the DH-RH reaction, as the breakdown peak was indeed observed in the HPLC analyses of these reaction mixtures.

The exact chemical structure of these oligomers could not be resolved due to the lack of purification methods that might have allowed for their evaluation by further analytical methods like MS-MS or NMR. Mass analysis of even the purified dimers was non-trivial due to the presence of excessive salt in the purified fraction, which resulted from the use of ion-exchange chromatography. Other chromatographic methods (such as ion-paired reverse phase chromatography) did not yield sufficient resolution for efficient purification. Nonetheless, these results strongly suggest that even in the reaction that contains base pairing nucleotides, loss of base continued to persist under our reaction conditions.

The reactions containing a mixture of all four nucleotides would, in principle, result in a complex mixture of oligomers that might be difficult to analyze. HPLC analysis of this reaction indicated the formation of oligomeric products similar to the ones seen in the binary combination of nucleotides, albeit with reduced yields [7]. Interestingly, the yields of the resultant oligomers seemed somewhat reduced in this reaction, as indicated by a lower peak intensity for the oligomers. This could potentially stem from the competition occurring between the monomers in these reactions. A breakdown peak was observed in this reaction as well, which most likely corresponded to purine bases that might have been lost due to cycling at low pH. The dimer peak resolved into multiple peaks, which could be attributed to high complexity in the dimer population resulting from the multiple combinations of interactions that are plausible between the four nucleotides. For example, there are at least 10 types of intact dimers that can form in the reaction independent of the order of the bases in them (viz. AA, AU, AG, AC, UU, UG, UC, GG, GC, CC). Apart from these, there would also be dimers that have one intact base and one abasic site (viz. Ar, Ur, Gr, Cr). MS characterization of this reaction, therefore, was found to be very challenging even at the level of the dimer population. This underscores the severe analytical constraints when working with multiple monomers under conditions that result in the preferential formation of specific products. Peaks observed in the spectrum had very low intensity/abundance, resulting in poor signal-to-noise ratios. This can partially also be attributed to poor ionization of the complex reaction mixture, thereby making the determination of the exact chemical species very difficult. Preliminary analysis indicated the formation of abasic oligomers similar to the ones that were observed in the A + U and G + C reactions. Nonetheless, further analysis could not be conducted due to the high ppm errors that were associated with the oligomers.

The difficulty in analyzing the products formed in the all four nucleotide reactions highlights the complexity that would have been intrinsic to a prebiotic reaction. Although polymerization of nucleotides ideally should take place independent of the nucleobase, this was not necessarily observed in any of our reactions. The polymerization of pyrimidines might have been slower due to the lack of efficient base stacking, and thus the formation of pyrimidine homopolymers might have been difficult in the presence of nucleotides containing other bases. Formation of abasic oligomers, which were predominantly detected in purine-only reactions, were also found to be the major products in this reaction. Importantly, the observed abasic oligomers would not be able to efficiently store information or transfer it, which diminishes their significance as prebiotically relevant informational polymers.

### 3.3. Deglycosylation Reactions during DH-RH Cycles

As both polymerization and deglycosylation reactions were being favored under similar conditions, we decided to further analyze the aspect of degradation of nucleotides under DH-RH conditions. The breakdown peak was predominantly observed in reactions involving purine nucleotides, either when used as monomers or even when present as mixtures. Since the AMP-based reactions always showed the most prominent breakdown, we studied the depurination aspect of this reaction in greater detail, i.e., at small time windows. The half-life (t_1/2_) of a glycosidic bond has been shown to decrease by about 5 times for dAMP with a concurrent increase in temperature from 37 °C to 50 °C [11]. Given this, it was estimated that for AMP, this might correspond to a decrease from a few days to a few hours or minutes when the deglycosylation reactions were to be conducted at 90 °C. Previous experiments have shown that DH-RH cycling was necessary for and facilitated the formation of RNA-like oligomers [3]. Therefore, to minimize the oligomerization in our reactions and focus on deglycosylation, for specific time intervals post-dehydration, the samples were analyzed without invoking rehydration. The samples were analyzed at 5, 10, 15, 30, 45, and 60 min post-dehydration (Figure 1). HPLC analysis showed that the breakdown peak was observed in as early as the 5 min sample, which indicated that loss of base took place very early on. However, unlike previous reports (e.g., Ref. [3]), a small amount of oligomerization was also observed in these samples even in the absence of DH-RH cycling.

Such oligomer peaks would interfere with the quantitation of deglycosylation, as some of the monomers (intact and otherwise) would also be utilized in the formation of oligomers. Furthermore, accurate quantification of oligomeric peaks could not be conducted due to the generation of abasic site(s) in the oligomers. Given these, deglycosylation was studied in the absence of polymerization by conducting the reactions in solution.

Oligomerization generally requires complete dehydration, as loss of water is not feasible under aqueous conditions. We, therefore, heated the AMP reaction mix at pH 2 and 90 °C under aqueous conditions in closed vials. Initially, time points as mentioned above were taken for this reaction as well; however, only about 6–7% depurination was observed in 1 h under aqueous conditions. This reaction was then followed for 7 h, which is equivalent to the duration of seven DH-RH cycles. In the pilot reaction, almost 50% depurination was observed in the 7 h sample and this was confirmed by repeating the reaction in triplicate. The average of percentage depurination (from three reaction replicates) was plotted versus time (Figure 2), wherein the time taken for degradation of 5′-AMP to half of the starting concentration was found to be ~6.35 h. This was about a 30-fold reduction in the t_1/2_ of AMP when compared to the previously reported t_1/2_ of 8.2 days that was obtained for reactions analyzed at pH 1 and 37 °C [11]. This rapid degradation of AMP at high temperatures poses a serious challenge to the feasibility of undertaking long-term reactions, as a large amount of AMP would be lost in a fairly short period of time. Similar experiments were also carried out with other canonical nucleotides; only 5′-GMP showed a high rate of breakdown that was similar to AMP. Both the pyrimidines viz. UMP and CMP did not show significant deglycosylation under similar reaction conditions, even after 7 h (Appendix A). This was in line with previously reported results, which showed that pyrimidines have greater *N*-glycosyl bond stability against purines [17,18]. Significantly, pyrimidines did not yield oligomers with an efficiency that was comparable to purines, which rather undermines their glycosyl bond stability in the context of the formation of informational molecules under prebiotic conditions.

## 4. Conclusions

Lipid-assisted polymerization was carried out with canonical nucleotides, as either individual monomers or binary mixtures capable of base pairing, or as a mixture of all four nucleotides. Oligomerization was observed in all the reactions, but the associated efficiency varied depending on the nature of the nucleobase. The results from MS analysis indicated the formation of abasic oligomers in almost all the reactions. Along with oligomers possessing abasic sites, intact dimers were observed only in pyrimidine-containing reactions. Purines underwent significant deglycosylation under the DH-RH conditions in all reactions studied. The extent of deglycosylation was also evaluated by characterizing the stability for purine ribonucleotides under the conditions used for DH-RH cycling. Specifically, under our DH-RH conditions, the apparent half-life of AMP was found to be only ~6.35 h, which was much lower than what has been previously reported, albeit under 37 °C [11]. Such high propensity for deglycosylation in purines raises imminent concerns about the stability of the glycosidic linkages in both monomers and the oligomers, with severe implications for storing and transferring information.

Results from this study, combined with previous relevant findings pertaining to a variety of issues surrounding canonical nucleosides [19,20,21,22], strongly advocate for the idea that modern nucleobases might have been preceded by different chemical ancestors. Specifically, low glycosidic bond stability in purines, under the same conditions that promote the formation of the phosphodiester bonds, indicates that the presence of both these moieties in the same monomer might have been unlikely under prebiotically pertinent conditions. Even though oligomerization involving non-activated nucleotides containing canonical bases may have been facilitated by other means [23,24], acid-catalyzed oligomerization would have not likely resulted in ‘intact’ informational polymers under prebiotic conditions.

Finally, mass analysis of complex reaction mixtures, such as the reaction containing all four nucleotides, is challenging at the very least. A study that reported MALDI analysis of oligomerization products using montmorillonite clay, had demonstrated the presence of up to 30-mer oligomers [25]. However, a common criticism has been the possibility of generating false positives during MS data acquisition, and hence there is a need for accurate calibration and careful sample preparation to minimize complications [26]. Few other studies have also resulted in a complex mix of products like the famous Formose reaction that presented major analytical challenges [27]. Nonetheless, despite the intrinsic difficulty associated with discerning such complex and heterogeneous mixtures, their analysis is very crucial for characterizing prebiotic reactions [28]. The contemplation of the inherently complex nature of prebiotic reactions and the efforts to simulate this and study them in detail has resulted in the recent emergence of the new field of ‘messy chemistry’ [29]. Nevertheless, it is also crucial to consider that the intrinsic heterogenic nature of the ‘substrate chemical space’ would potentially lead to an even more complex and diverse ‘product chemical space’. More importantly, attempts to analyze the reactions with mixtures of starting material, which represent the complexity associated with prebiotic inventory, have recently begun [30]. Nonetheless, development of highly sensitive and robust analytical techniques, as well as bold approaches towards simulating and characterizing complex prebiotic reactions, is increasingly being acknowledged as being fundamental to solving the mystery of the origins of life on Earth.

## Figures and Tables

**Figure 1 life-09-00057-f001:**
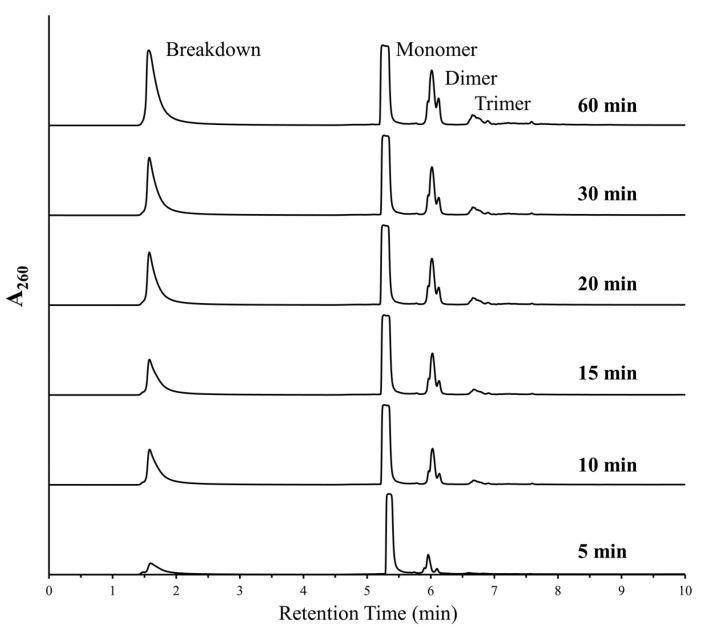
Breakdown and marginal polymerization was observed in reactions where AMP was heated at 90 °C, pH 2, without rehydration. HPLC analysis showed increasing peaks for free nucleobases (breakdown) and higher oligomers (predominantly dimers and trimers) with increased duration. This suggests that the cleavage of the glycosidic bond and oligomerization both occur under simultaneous experimental conditions.

**Figure 2 life-09-00057-f002:**
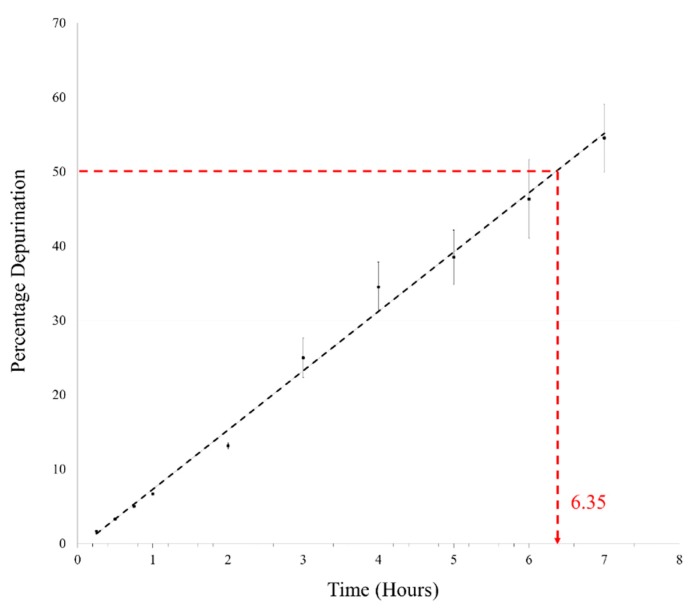
Percentage depurination is plotted against time for estimating the degradation of AMP under reaction conditions used for dehydration-rehydration (DH-RH) reactions. AMP gets degraded to about half of the starting concentration in ~6.35 h at pH 2 and 90 °C. (n = 3, R^2^ = 0.9941).

**Table 1 life-09-00057-t001:** Mass numbers observed in the reactions containing individual nucleotides. Detailed mass spectra and peaks obtained for the individual nucleotide polymerization reactions are included in the Appendix A.

Chemical Species	Expected Mass	Observed Mass	ppm Error
**AMP Reaction**			
Adenine	136.0617	136.0635	13.2290
AMP monomer	348.0703	348.0691	3.4475
Abasic Dimer	560.0778	560.0764	2.4996
Abasic Trimer	772.0874	772.0899	3.2379
**GMP Reaction**			
Guanine	152.0566	152.057	2.6305
GMP Monomer	364.0652	364.0627	6.8669
Abasic Dimer	576.0737	576.0715	3.8189
Abasic Trimer	788.0824	788.0803	2.6646
**UMP Reaction**			
Uracil	113.0345	113.0346	0.8846
UMP Monomer	325.0431	325.0435	1.2306
Intact Dimer	631.0684	631.0682	0.3169
Abasic Dimer	537.0517	537.0533	2.9792
**CMP Reaction**			
Cytosine	112.0505	112.0494	9.8170
CMP Monomer	324.0591	324.0596	1.5429
Intact Dimer	629.1004	629.1017	2.0664
Abasic Dimer	536.0677	536.0677	0.0000

**Table 2 life-09-00057-t002:** Peaks observed in MS analysis of mixed nucleotide reactions. Potential structures for some of the chemical species is depicted in Appendix A. Detailed mass spectra and peaks obtained for reactions containing base pairing nucleotides are included in the Appendix A.

Chemical Species	Expected Mass	Observed Mass	ppm Error
**AMP + UMP Reaction**			
Adenine	136.0617	136.0635	13.2290
AMP Monomer	348.0703	348.0691	3.4475
Uracil	113.0345	113.0346	0.8846
UMP Monomer	325.0431	325.0435	1.2306
Abasic A Dimer	560.0778	560.0812	6.0705
Abasic U Dimer	537.0517	537.0533	2.9792
Abasic A Trimer	772.0874	772.0899	3.2379
Abasic U Trimer	749.0603	749.0637	4.5390
**GMP + CMP Reaction**			
Guanine	152.0566	152.057	2.6305
GMP Monomer	364.0652	364.0627	6.8669
Cytosine	112.0505	112.0494	9.817
CMP Monomer	324.0591	324.0596	1.5429
Intact CC Dimer	629.1004	629.1017	2.0664
Intact CG Dimer	669.1065	669.1042	3.4374
Abasic G Dimer	576.0737	576.0715	3.8189
Abasic C Dimer	536.0677	536.0677	0.0000
Abasic G Trimer	788.0824	788.0803	2.6646
Abasic C Trimer	748.0763	748.0787	3.2082

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
