# Peer review of "Formation of Abasic Oligomers in Nonenzymatic Polymerization of Canonical Nucleotides"

_life, 2019, doi:10.3390/life9030057_

Reviewer 1 Report

This article deal with the polymerization of non-activated nucleotides under prebiotically plausible conditions that is a crucial event in early evolution of life. Several questions remain in particular the depurination events at low pH (and dry-wet condition) resulting in abasic sites in the oligomers with dramatic consequences for base pairing mandatory for replication.

Many different conditions are explored that show that the RNA is more stable than DNA. In addition, AMP is the most stable nucleotide.
It would be desirable for Figure 1, although correctly commented in the text, to have a more developed legend.
On the other hand, the peaks in Figure 1 must be identified in the figure itself.

The experimental methods used are fine and some conclusions are thought-provoking in the field of Orifins of life study.

Finally, in line 300 add an S at 7 hours.

After these few corrections I recommend the publication of this article

Reviewer 2 Report

Mungi et al. report the characterisation of acid-catalysed nucleoside oligomerization using dry-wet cycles in the presence of a lipid. They report depurination and depyrimidination of short oligonucleotides synthesised under those conditions, consistent with current knowledge. The mixed-based aspect of the work is interesting but as the authors themselves state, difficult to deconvolute and conclusions are of limited value.

My comments are roughly in the order they appear in the manuscript. Particularly relevant comments are highlighted with *.

1.       There are some small grammatical and orthographic mistakes in the manuscript. They generally don’t interfere with understanding. On L. 48, however, the mistake does interfere with meaning and it is not clear if abasic RNA is more, or less stable than abasic DNA.

2.       L.62-63 ‘As seen in aforementioned… RNA’. There is no need for this repetition given the content of the preceding paragraph.

3.       *Intro (L.86-93) – I think language here should be more guarded. Wet-dry cycles reaching 90oC in a highly acidic environment are not ‘realistic’. Notwithstanding that synthesis and degradation may be affected differently by changes in temperature, there are now multiple scenarios (see work by Sutherland, Holliger or Powner) suggesting that different routes are viable and could even be carried out in ice. It may be fair to say that prebiotic conditions limited to acidic environments and high-temperature wet-dry cycles are not compatible with the early emergence of DNA and RNA.

4.       L.125 ‘This was done in order… conditions’ I do not understand that sentence. Does it mean that those conditions were used so that the extent of oligomerization caused by the acid bath alone could be measured? Or so that no oligomerization could occur?

5.       In line with the previous comment, it is not clear from the manuscript if there were qualitative or quantitative differences in the synthesised oligonucleotides seen between the single wet oligomerization cycle and the 9-cycle approach. That is a crucial issue if an argument is being made that the wet-dry cycles are essential in the reaction.

6.       L.148 Yayyoi Hongo is an author in the paper, thus the opening sentence in this section is unnecessary.

7.       It may be useful to link Table 2 to a figure showing at least some example molecules that are being described as Abasic A dimer, abasic U dimer and higher. It will make the article more accessible to researchers in other fields and would make it easier to highlight the ‘loss of information’ aspect of the reaction.

8.       *Deglycosylation in DH-RH cycling. The graph suggests deglycosylation as a zeroth-order reaction, which is unexpected. Could the authors provide more explanation on the choice of fit here and discuss it in light of the what would be expected (i.e. first order reaction possibly with some linear displacement due to method)?

9.       *L.337 – Following on from point 3 above, I think a more guarded language is needed here since there is little evidence that the reaction conditions being proposed are likely to have been found prebiotically. There are now well-established system chemistry results that propose viable routes towards the synthesis of the natural nucleobases and close analogues, so the need to argue for a pre-RNA world would go unsupported.
